# Networked Learning Communities in Promoting Teachers' Receptivity to Change: How Professional Learning Beliefs and Behaviors Mediate

**Hui-Ling Wendy Pan [1]** and **Wen-Yan Chen [2,\*]**

1   Department of Education and Future Design, Tamkang University, New Taipei 251301, Taiwan
2   Department of Educational Policy and Administration, National Chi Nan University, Nantou 545301, Taiwan
\*   Correspondence: wychen@ncnu.edu.tw

**Abstract:** More research on networked learning communities (NLCs) and the causal mechanism among the effects of NLCs are needed. To better understand the impacts of NLCs and the influential factors, this study intended to discover how teachers' participation in networked learning communities affects their beliefs and behaviors of professional learning and further influence their receptivity to change. Adopting a survey design, we collected 226 valid questionnaires from the pilot schools joining the program of Learning Community under Leadership for Learning supported by the Ministry of Education in Taiwan. First, the results indicated that the program's intervention of NLCs had a significant positive effect on teachers' receptivity to change. Second, teachers' participation in NLCs also showed a significant impact on their beliefs and behaviors regarding professional learning. Third, teachers' beliefs in professional collaborative learning could significantly enhance their behaviors of professional collaborative learning. Fourth, the program's intervention, employing hands-on professional learning activities, had a significant impact on teachers' inclination to realize the program, both through direct influence and the mediation of beliefs about professional learning. The results acquired from this study would be conducive to developing strategies to support implementing the NLCs program.

**Keywords:** curriculum reform; lesson study; networked learning community; professional learning community; program evaluation; receptivity to change; teacher beliefs; teacher behaviors; teacher change; teacher learning



## 1. Introduction

Responding to the challenges of a fast-changing society, many countries in the 21st century have launched curriculum reform to equip students with competencies for the future [1]. A constructivist-based curriculum features this wave of reform [2]. It demands that teachers transform their roles from delivering knowledge to facilitating student learning. However, receptivity to change is a prerequisite for teachers to move toward learner-centered pedagogy. To trigger the drive to innovate teaching, teachers can collaborate to create new professional knowledge, make professional experiences visible, and share instructional practices.

A variety of professional activities provides teachers to collaborate. One of the options is professional learning communities (PLCs). In the PLCs, teachers share ideas, resources, and expertise to improve their professional practice and students' academic success. Noteworthily, learning communities do not only occur within a school. A recent focus is on working across schools [3]. This form of teacher collaboration is referred to as networked learning communities (NLCs). Clusters of schools work in partnership to uplift professional learning quality and engage in continuous improvement [4]. Whether PLCs or NLCs, many countries have promoted them as a promising approach to school improvement. Since

2010, in Taiwan, the Ministry of Education has also taken steps to encourage their development [5]. In addition, a different approach to learning communities developed by Manabu Sato, a Japanese scholar, was introduced to Taiwan [6,7]. Pan et al. [8,9] also developed an indigenous model of learning communities to meet the specific needs of teachers in the country and embarked on a learning community program across cities/counties. The program provides opportunities for teachers to work across schools. Investigating its effects renders this study.

The previous studies revealed that networked learning communities could be instrumental in helping teachers work across schools and share knowledge and resources [3,4,10,11]. By collaborating with teachers from other schools, teachers in NLCs can gain a broader perspective on teaching and learning and access a more comprehensive range of ideas and strategies. It can be constructive in addressing the increased complexity of challenges teachers face. Nevertheless, the effects of NLCs are not yet conclusive [12,13]. How learning communities result in the desired outcomes and if teachers' beliefs and behaviors play a role in the causal mechanism also await to be understood. Moreover, receptivity to change is influential in implementing reform initiatives [14,15]. Investigating the concept of receptivity may provide hints for policymakers and school leaders to design effective policy tools. So, we used it as the outcome variable. In summary, this study aimed to examine the program effect of NLCs in promoting teachers' inclination to adopt the change initiatives regarding operating their classrooms as learning communities and being actively involved in teacher learning communities. A further attempt was to unpack the causal mechanism among teachers' participation in NLCs, beliefs and behaviors of professional learning, and receptivity to change.

## 2. Conceptual Background

### 2.1. Networked Learning Communities

Professional learning communities are one of the most effective strategies for teacher change and school improvement [16–20]. They involve a paradigm shift from traditional professional development focusing on one-shot activities to continuous learning in the workplace [21–23]. In the literature, PLCs are described with the characteristics of shared values and vision, collective responsibility, reflective professional inquiry, and the promotion of group and individual learning [24,25]. In addition to the discourses and practices of PLCs developed by the West, there are also unique ways created in Asia to implement PLCs. In Japan, Sato [6,7], based on the theories of Dewey and Vygotsky, proposes the "learning community" (xue xi gong tong ti) approach to transform schools. Building collegiality among teachers and constructing classrooms as learning communities through collaborative learning are the two essential elements of his viewpoints. The former originates from traditional lesson study in Japan. As a Japanese form of professional development, lesson study refers to the collaborative study of classroom lessons [26]. Teachers conduct an action-inquiry cycle in three steps: working together to plan the lesson, conducting the lesson with one teacher teaching and others observing, and discussing the lesson taught based on the data collected [27].

In a reform context of extending basic education from nine to twelve years, Sato's learning community approach was introduced to Taiwan in 2012 [7]. To accommodate the needs of local teachers, Pan and colleagues [8,9] constructed an indigenous learning community model called *Learning Community under Leadership for Learning*. It integrates the conceptualizations from Sato, Western theories, and place-based discourses and practices. The university, school, and government partnership was established to promote the indigenous learning community model. There were schools from five cities/counties which joined the program. The program's operation has made PLCs not limited to one school. Networked learning exists across schools.

Based on the theory of action, professional learning materializes significant changes in practice. In the learning process, interactions occur within and across schools. Networked learning communities distinguish themselves from other networks by focusing on learning.

Their orientation is to involve collaborative participation in creating knowledge. Through the process of engaging in cultural practices and shared learning activities, individual and collective knowledge in the communities is enriched or transformed. The creation of knowledge leads to participants' cognitive reframing and new ways of working that can be applied to classrooms and schools [4,28]. The networked learning communities are to nurture innovative knowledge communities within schools by connecting school-based groups with their counterparts in other schools. They can facilitate the key actors to share ideas and practices and create new knowledge. Individuals play a role in connecting schools and networks through active participation in the community and the construction of artifacts that serve as links between the network and the school. It creates a two-way flow of information and knowledge, allowing schools to upload and download ideas and practices from the network [3,4,17].

### 2.2. Professional Learning

Professional learning activities are designed to advance teachers' knowledge and skills and develop new teaching approaches that can help improve student achievement. However, traditional professional development strategies have been criticized as being fragmented, poorly conducted, and neglectful of the role of adult learning [29,30]. Cole [31] also commented that conventional teacher training was often decontextualized from teachers' classroom practice. Professional development, to be effective, requires teachers to manifest their learning processes in their roles in the classrooms and school communities [30].

From a complexity theory perspective, teaching knowledge is not a static body of information in the teacher or outside of the teacher [32]. Instead, it is a dynamic and constantly evolving process influenced by various factors, such as the teacher's experiences and interactions with colleagues and students. In this view, learning is also a continuous process that involves the ongoing transformation of both the learner (teacher) and the knowledge being learned. As the teacher learns and gains new knowledge, their understanding of the subject matter and teaching practice are transformed. Thus, more intensive, collaborative, job-embedded, and long-term professional learning is needed, which is PLCs [4,22,33,34].

Lesson study is another approach to professional learning that has been successful in Japan and other Asian countries. In lesson study, teachers work together to plan and observe lessons and then reflect on the results to improve instruction. Another model for teacher learning is the Taiwanese model, which promotes learning communities at the classroom, teacher, and school levels. As noted earlier, Sato's [7] learning community approach was transformed with other ingredients into an ingenious Taiwanese model [8,9]. In this model, students, teachers, and staff can learn and grow together when shaping the school as a learning community. Teacher learning communities gather teachers to share knowledge and expertise, ask questions, and seek feedback from colleagues. In the classrooms as learning communities, teachers enact learner-centered pedagogy, students learn collaboratively, and all the participants work together and support one another. As a result, learning occurs between teachers, between teachers and students, and between students.

There have been numerous studies on professional learning communities (PLCs) that have examined their effects on various aspects of education, including teachers' instructional practice [35,36], teacher trust and commitment [14], and student learning outcomes [37]. However, there is limited research on the causal linkage of learning community effects [3,34], and the evidence for the effectiveness of NLCs is inconclusive [12,13].

Another area that has received little attention in the research on learning communities is the way in which teachers' beliefs and behaviors may change as a result of participating in networked learning communities. This is a critical issue to explore as understanding how teachers' beliefs and behaviors are impacted by their participation in NLCs can help us understand how they may be more receptive to change. The previous research on beliefs and behaviors in education has often focused on teaching and learning [38–42]. This study attempted to fill the gap by examining how teachers' beliefs and behaviors

about professional learning may mediate the effect of participation in NLCs on receptivity to change.

### 2.3. Receptivity to Change

Receptivity to change refers to an individual's inclination to adopt new ideas, practices, or processes. In the literature, receptivity is often measured using four aspects: "character-istics of the change, managing the change at school, value for the teacher, and perceived value for students" [43] p. 358. Behavioral intention is employed as an indicator to assess teachers' receptivity toward reform initiatives [43–45]. Several studies applying the model proposed by Waugh and colleagues [42–44] have explored teacher receptivity to reform in Hong Kong and China to understand how it influences teachers' implementation of change initiatives [14,15].

The measurement of teacher receptivity considers the content of the proposed change initiatives. So, we need to address the reform context of the research site. In Taiwan, the government has encouraged the adoption of constructivist pedagogy and collabora-tive curriculum development as part of an effort to extend basic education from nine to 12 years and implement the new Curriculum Guidelines [46]. In this study, we examined teachers' receptivity to change regarding implementing the learning community program, which aligns with the reform trend of promoting learner-centered pedagogy and teacher collaboration. Understanding teachers' receptivity to this program can help policymak-ers and school leaders support teachers in implementing the program and achieving the desired outcomes.

As noted earlier, investigating receptivity to change has practical implications for program implementation, and the reviewed literature unfolds that explorations of the causal mechanism for networked learning communities are still scarce. Therefore, we intended to discover how teachers' participation in NLCs affects their beliefs and behaviors of professional learning and further influences their receptivity to change. The conceptual framework proposed is displayed in Figure 1. The research questions we formulated are as follows:

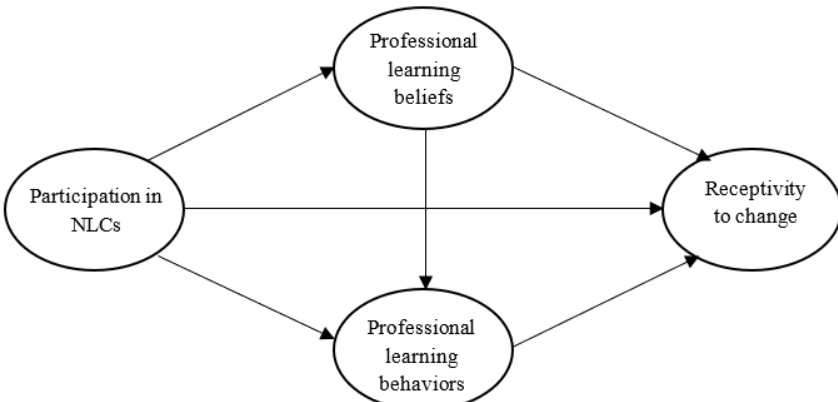

**Figure 1.** The conceptual framework.

1. What are teacher perceptions of participation in networked learning communities, beliefs and behaviors of professional learning, and receptivity to change?

2. How does teachers' participation in networked learning communities affect beliefs and behaviors of professional learning and receptivity to change?

3. What are the associations between teachers' beliefs and behaviors of professional learning and their receptivity to change?

## 3. Methodology

### 3.1. Participants and Procedures

This study employed a quantitative survey design to examine the effects of the networked learning community program. The participants were selected from the pilot schools of the *Learning Community under Leadership for Learning* program supported by the Ministry of Education in Taiwan. There were 737 teachers at the 33 pilot schools (15 elementary, 16 junior high schools, and two senior high schools). This study only analyzed the data of elementary and junior high school teachers. With half of the teachers participating in the program as subjects (including teachers, office directors, and office section chiefs), a total of 226 valid questionnaires were collected from an online survey. The characteristics of the respondents are shown in Table 1.

**Table 1.** Characteristics of the respondents.

| Variables | Categories | N | % |
|---|---|---|---|
| Gender | Male | 75 | 33.2 |
| | Female | 151 | 66.8 |
| Years of experience at the school | 5 years and below | 66 | 29.2 |
| | 6–10 years | 35 | 15.5 |
| | 11–15 years | 55 | 24.3 |
| | 16–20 years | 43 | 19.0 |
| | 21 years and above | 27 | 11.9 |
| Duty | Office directors | 39 | 17.2 |
| | Office section chiefs | 46 | 20.3 |
| | Homeroom teachers | 101 | 44.7 |
| | Subject teachers | 40 | 17.7 |
| Subjects | Mandarin | 83 | 36.7 |
| | English | 26 | 11.5 |
| | Mathematics | 19 | 8.4 |
| | Natural sciences | 30 | 13.3 |
| | Health and physical education | 15 | 6.6 |
| | Social studies | 13 | 5.8 |
| | Integrative activities | 11 | 4.9 |
| | Arts | 17 | 7.5 |
| | Life curriculum | 1 | 0.4 |
| | Other types of courses | 11 | 4.9 |
| School level | Elementary school | 131 | 58.0 |
| | Junior high school | 95 | 42.0 |

### 3.2. Instruments

In order to measure the effects of the networked learning community program, the questionnaire consisted of four scales. Teacher participation in the learning communities is conceptualized as the program activity (independent variable). Professional learning beliefs and behaviors are regarded as the short-term program outcomes (mediating variables), and teachers' receptivity to change is used as the intermediate program outcome (dependent variable). To confirm the construct validity of the scales, we conducted confirmatory factor analysis (CFA) and reported values of the composite reliability (CR) and average variance extracted (AVE). CR shows the degree of internal consistency of the latent variables, with a value higher than 0.60 as the standard [47]. AVE indicates the average variation explanatory power of each observed variable to the latent variable to which it belongs, and the value is preferably higher than 0.50 [48].

*Networked learning communities.* Teachers' participation in learning communities is used to assess their critical experiences of the program. The scale consists of three items: "participating in class observations in learning communities", "participating in discussion

after class observation in learning communities", and "participating in joint lesson planning in learning communities". On a five-point Likert-type scale, participants were asked to respond to the frequencies of their participation in each activity from "never", "one to two", "three to four", "five to six" to "seven and more" times. The CR value was 0.93, and the AVE value was 0.81.

*Professional learning beliefs.* Based on social constructivism [49] and the ideas of teacher learning communities proposed by Sato [7] (2012) and Pan et al. [8,9], three items were developed to measure teachers' beliefs of professional learning. Participants were asked to rate their actual feelings about the statement of each item on a six-point Likert-type scale (1 = strongly disagree; 6 = strongly agree). The items are "Although joint lesson planning takes more time, it is more effective and rewarding than doing it alone", "Although the open class is a bit disturbing, it is still worth it", and "As a teacher, you must be able to discuss your teaching ideas and methods with others in order to teach better". The CR value was 0.76, and the AVE value was 0.52.

*Professional learning behaviors.* Teachers' behaviors of professional learning are evaluated by three items using a six-point scale. These items reflect the behaviors that are encouraged in the communities [5,9]. The items are "I discuss with my peers how to design learning activities, such as big ideas, key questions, and what students are able to know and do", "I discuss with my peers whether and where student learning is happening", and "I discuss the multifaceted nature and particularity of student learning with peers through class observation". The CR value was 0.88, and the AVE value was 0.71.

*Receptivity to change.* In order to evaluate teachers' receptivity to implement the learning community program, three items are developed using a six-point scale. The items are "I like to teach using the learning community model", "I am proud of being a teacher who implements the learning community model", and "I like to think about how to teach better by using the learning community model". The CR value was 0.93, and the AVE value was 0.82.

Overall, the CR values for the four scales ranged from 0.71 to 0.93, which all exceeded the desired level of 0.60. The AVE values for the four latent variables also met the required standard (>0.50) [48]. The indexes indicate that the model's convergent validity was satisfactory. Concerning the discriminant validity, it was verified by comparing the square root of the AVE of each variable with the correlation coefficients of the variable with other variables. A variable is regarded as distinctive from other variables when the square root value is higher than the correlation coefficient [48]. In Table 2, the square root of the AVE of each variable in the diagonal is greater than its contrasting correlation coefficients. Therefore, it shows acceptable discriminant validity of the model.

**Table 2.** Discriminant validity of the main constructs.

| | AVE | Participating in NLCs | Professional Learning Beliefs | Professional Learning Behaviors | Receptivity |
|---|---|---|---|---|---|
| Participation in NLCs | 0.81 | **0.90** | | | |
| Professional learning beliefs | 0.52 | 0.18 | **0.72** | | |
| Professional learning behaviors | 0.71 | 0.29 | 0.71 | **0.84** | |
| Receptivity to change | 0.82 | 0.34 | 0.67 | 0.58 | **0.91** |

### 3.3. Analysis Strategies

We used SPSS 24 and AMOS 24.0 to conduct the statistical analysis. First, the descriptive statistics, including means, standard deviations, and correlations of the variables, were computed to understand teachers' participation in networked learning communities, their beliefs and behaviors of professional learning, and their receptivity to change. Second, structural equation modeling was conducted to verify the relationships among the variables. Generally recommended indices, including the root mean square error of approximation (RMSEA), the comparative fit index (CFI), the Tracker–Lewis index (TLI), and the standardized root mean squared residual (SRMR), were used to determine the model fit. The

standard values CFI $\geq$ 0.90, TLI $\geq$ 0.90, RMSEA $\leq$ 0.08, and SRMR $\leq$ 0.08 were used as cutoffs for acceptable data fit [50]. Third, to ensure the quality of the mediation analysis, bootstrapping was used to confirm the significance of the indirect effect by resampling the data 5000 times to yield a parameter estimate for indirect and total effects. When the 95% bias-corrected confidence interval for the parameter estimate does not contain zero, the mediating effect is regarded as statistically significant [51,52].

## 4. Findings

### 4.1. Descriptive Statistics and Correlation Analysis

The descriptive statistics of all variables are presented in Table 3. The mean score for teacher participation in networked learning communities was 2.74 on the five-point Likert-type scale. Regarding beliefs and behaviors of professional learning, the mean scores were 4.71 and 4.74 on the six-point scale, respectively, which were at a high–intermediate level. In terms of receptivity to change, the mean score was 4.46. It shows that teachers had a moderately high willingness to participate in the program. Concerning the correlation, the four variables were all positively related.

**Table 3.** The means and correlation matrix.

| Variables | M | SD | 1 | 2 | 3 |
|---|---|---|---|---|---|
| 1. Participation in NLCs | 2.74 | 0.96 | | | |
| 2. Professional learning beliefs | 4.71 | 0.72 | 0.18 ** | | |
| 3. Professional learning behaviors | 4.74 | 0.76 | 0.31 *** | 0.59 *** | |
| 4. Receptivity to change | 4.46 | 0.91 | 0.36 *** | 0.57 *** | 0.54 *** |

Note: ** refers to $p < 0.01$, *** refers to $p < 0.001$.

### 4.2. The Effects of Networked Learning Communities on Professional Learning and Receptivity to Change

Based on the theory of the teacher learning community program, the study hypothesized the relationships among teacher participation in networked learning communities, beliefs and behaviors of professional learning, and receptivity to change and examined them using a mediation model. The standardized estimation of the structure model is shown in Figure 2, with a satisfactory fit for the data (RMSEA = 0.040, CFI = 0.991, TLI = 0.988, SRMR = 0.041) [47,53,54].

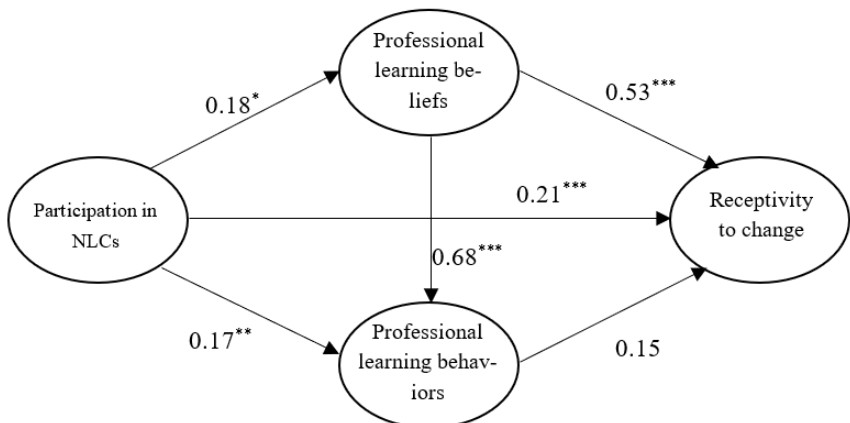

**Figure 2.** The mediation model of networked learning communities affecting professional learning and receptivity to change. Note: * refers to $p < 0.05$, ** refers to $p < 0.01$, *** refers to $p < 0.001$.

Examining the influencing paths, the program activity, namely, teachers participating in networked learning communities, could directly affect teachers' beliefs of professional learning ($\beta = 0.18$, $p < 0.05$), behaviors of professional learning ($\beta = 0.17$, $p < 0.01$), and receptivity to change ($\beta = 0.21$, $p < 0.001$). Teachers' beliefs in professional learning had

a significant positive effect both on their behaviors of professional learning (β = 0.68, $p < 0.001$) and receptivity to change (β = 0.53, $p < 0.001$). However, teachers' behaviors of professional learning did not have a significant impact on their receptivity to change (β = 0.15, $p > 0.05$).

To confirm the impact of the mediation effects, we conducted significance tests on the mediators' specific indirect effects using bootstrapping procedures with a 95% percentile interval. The results in Table 4 reveal that participating in networked learning communities could promote teachers' receptivity to change by strengthening teachers' beliefs in professional learning. However, the impact of participating in networked learning communities on receptivity to change was not effective through the mediation of behaviors of networked learning communities. In other words, the model displayed a partial mediation, and teachers' beliefs in professional learning were the key mediator.

**Table 4.** Bootstrapping results of standardized indirect effects.

| | Point Estimates | Product of Coefficients | | Bootstrapping | | |
|---|---|---|---|---|---|---|
| | | | | Percentile 95% CI | | |
| | | SE | Z | Lower | Upper | *p* |
| Participation in NLCs→PL Beliefs→Receptivity | 0.089 | 0.044 | 2.023 | 0.017 | 0.194 | 0.015 |
| Participation in NLCs→PL Beliefs→Receptivity | 0.023 | 0.023 | 1.000 | −0.010 | 0.086 | 0.159 |
| Participation in NLCs→PL Beliefs→PL behaviors→Receptivity | 0.017 | 0.019 | 0.895 | −0.007 | 0.076 | 0.176 |

Note: PL Beliefs: professional learning beliefs, PL behaviors: professional learning behaviors.

## 5. Discussion and Conclusions

In order to assist teachers in operating their classrooms as learning communities and enacting learner-centered teaching, an essential part of the *Learning Community under Leadership for Learning* program investigated in this study is the teacher learning community of NLCs. It encourages teachers to participate in lesson study for sustainable professional development and support developing learner-centered classrooms [8,9,55]. By being exposed to the practice of lesson study, teachers can experience and practice the process of this "new" approach to professional learning. We gathered data from elementary and junior high school teachers in Taiwan and examined the extent to which participation in NLCs leads to changes in teachers' beliefs and behaviors of professional learning and receptivity to change. In addition, we addressed the research gap by unpacking the causal mechanism of the NLCs' effects. Four main findings can be concluded as follows:

First, the program intervention of networked learning communities had a significant positive effect on teachers' receptivity to change. Second, teachers' participation in networked learning communities also showed a significant impact on their beliefs and behaviors regarding professional learning. Third, teachers' beliefs in professional collaborative learning could significantly enhance their behaviors of professional collaborative learning. Fourth, the partially-mediated model showed that the program intervention employing hands-on professional learning activities had a significant impact on teachers' inclination to realize the program, both through direct influence and the mediation of beliefs about professional learning. In other words, although teachers' beliefs and behaviors about professional learning were important factors that affected their receptivity to the new program, only beliefs had a statistically significant mediating effect.

Three meaningful issues surfacing from the findings are worth discussing. The first is about the effects of networked learning communities. As the networked learning communities program promotes a form of contextualized collaborative learning rooted in authentic classroom situations, teachers can develop and deepen their knowledge through interac-

tions in the community of practice [56]. In particular, teachers participating in the program are encouraged to operate the lesson study procedure. Teachers can learn how to cultivate a learner-centered classroom through observation. They also learn how to plan a lesson collectively, how to observe a class, give feedback based on observations, and understand the whole procedure by joining the lesson study activities provided by the program. In addition, teachers not only exchange ideas within the school but also observe, discuss, or plan lessons with colleagues from other schools. Teachers actively engaged in their learning and are open to new ideas and approaches, particularly acquired benefits from external professional advice and resources through the NLCs. Teachers enhance their risk-taking skills and adapt to innovative teaching approaches through collaboration [57,58], contributing to their inclination to change.

When teachers witness the feasibility of classroom practice and the changes in student learning, their beliefs and behaviors may change with it [59,60]. Individual and collective knowledge is enriched in the learning process. The new knowledge teachers create leads to the reframing of conceptualization and finding new teaching approaches [4,28]. A two-way flow of information and knowledge further affords schools to upload and download thoughts and experiences from the network [3,4,17]. It broadens the avenues to promote the possibilities to render changes in teachers' beliefs and practices. Teachers' professional identities may also be reshaped gradually [61,62]. In sum, the results of this study demonstrated the effects and value of experiential learning in networked learning communities.

The second issue is about the relationship between beliefs and behaviors. There are different views in the previous literature on which of the changes comes first. Early theories of change derived from Lewin's [63] psychotherapeutic model held that professional development activities lead to changes in beliefs and attitudes, followed by changes in classroom behavior. However, some other studies argued that teachers only developed their recognition when they used it in the classroom. Guskey [64] proposed a new interpretation model by advocating that teachers' teaching behaviors change first. Then comes the change in teachers' beliefs and attitudes as teachers witness the improvement in students' learning.

Third, our conceptual framework hypothesized that teachers' participation in networked learning communities could directly affect teachers' receptivity and indirectly through its impact on teachers' beliefs and behaviors. However, the findings only suggested teacher beliefs as an effective mediator. It reveals that in practice-oriented learning communities, teachers could observe, participate, and witness the changes brought by practices. As a result, their temptation to experiment with operating their classrooms and active engagement in professional collaboration is triggered. Therefore, changes in teachers' beliefs were sufficient to enhance their receptivity to the program rather than through the mediation of behavioral change.

The findings highlight the importance of providing teachers with opportunities to engage with new ideas and practices in a supportive environment. By participating in networked learning communities, teachers can learn from each other and the experiences of their colleagues, which can help to shape their beliefs about teaching and learning. It, in turn, can increase their receptivity to change and encourage them to try new approaches in their classrooms. So, first of all, it is suggested to utilize practice-oriented teacher collaborative learning as the intervention in teacher learning and instructional innovation programs. It is more authentic and convincing for teachers to learn to change in a real classroom context and by observing, discussing, and practicing. Second, from this study, policymaking and school leaders may realize that teachers' beliefs are crucial for their willingness and openness to adopting new initiatives. Understanding and assessing teachers' beliefs would be essential when implementing reforms. Finally, the direction for future research is worthy of being noted. The study utilized a survey design to verify the influencing paths of an NLCs program. The more nuanced reasons why these effects take place is a question that further exploration could address. As demographic and personal variances of teachers might influence teachers' beliefs and tendency to change and affect the results of the survey,

comparing the variabilities can also provide insightful findings. Moreover, the effects of teacher NLCs on student learning is another critical empirical issue to be explored.

**Author Contributions:** Formal analysis, H.-L.W.P.; Investigation, H.-L.W.P.; Writing—original draft, H.-L.W.P. and W.-Y.C.; Writing—review & editing, H.-L.W.P.; Visualization, H.-L.W.P. All authors have read and agreed to the published version of the manuscript.

**Funding:** This research was financially supported by the Ministry of Education of Taiwan (grant no: 1036004).

**Institutional Review Board Statement:** Not applicable.

**Informed Consent Statement:** Not applicable.

**Data Availability Statement:** Data is unavailable due to privacy or ethical restrictions.

**Conflicts of Interest:** The authors declare no conflict of interest.

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
