# Peer review of "Networked Learning Communities in Promoting Teachers’ Receptivity to Change: How Professional Learning Beliefs and Behaviors Mediate"

_sustainability, doi:10.3390/su15032396_

Round 1

Reviewer 1 Report

The paper is well-written, but the originality isn't very high.

Authors research how teachers' participation in networked learning communities affects their beliefs and behaviors of professional learning and further influence their receptivity to change. 

The manuscript is presented in a well-structured manner. The authors use an appropriate experimental design. The figures and tables are clear and illustrate the presented results.    

I have some minor observations:

* I recommend publishing the questionnaire with which the survey was conducted.

* Аdd more references from the last 5 years.

* The quoted source is not specified in several places in the text. For example line 45 (Author, 2014), line 92 (Author and colleagues (2014, 2015)), line 329 (Author et al., 2014, 2015), and others. The References list has the same problem - ref. 3, 4, 5, 6.

* There are minor grammatical errors. The paper should be proofread.

Reviewer 2 Report

Outstanding rigor and explanation of the applied structural equation model. The analysis is preceded by a clear delineation of the conceptual framework as well as the need for ongoing research in this topic area of networked learning communities and change. Implications for professional practice and application by teachers are identified. I would recommend adding even one or two more recommendations for future related research to keep building upon the findings of the study. I would add one more word to the title: "Mediate....(what? missing noun?)" Overall an excellent analysis worthy of publication.

Reviewer 3 Report

1. The research is well designed and the conclusion is worth to extend.  I would ask if there is variability between survey results from junior school teachers and elementary teachers? Since teachers' professional identities may not be consistent between levels. 

2. The learning community model is rewarding and conductive, but how to implement the learning community model should be described.

3. How about the students achievement involved in the survey? Should  the survey expand to students be more objective?
